# Capital Structure and Its Determinants—A Comparison of European Top-Rated CSR and Other Companies

**Peter Krištofík \*** , **Juraj Medzihorský** and **Hussam Musa**

Faculty of Economics, Matej Bel University, Tajovského 10, 975 90 Banská Bystrica, Slovakia; juraj.medzihorsky@umb.sk (J.M.); hussam.musa@umb.sk (H.M.)
\* Correspondence: peter.kristofik@umb.sk; Tel.: +421-48-446-2111

**Abstract:** Corporate social responsibility (CSR), ethics, and sustainability have become an inseparable part of the discourse of modern business. Applying linear regression and comparison of intervals of beta-coefficients, we focused on the mediating role of CSR in the relations between capital structure and its determinants. Examining the sample of European large caps, we observed that CSR companies are significantly more leveraged than non-CSR ones. The influence of the corporate income tax rate and depreciation and amortization on leverage does not differ significantly between CSR and non-CSR companies. Moreover, tax shields seem to be insignificant for both CSR and non-CSR companies. However, we should stress that, for depreciation and amortization, the beta coefficient has a different significance in the model of CSR companies, compared to the model of non-CSR companies. Also, the difference between the models regarding the relations of leverage and asset tangibility is worth noting. Non-CSR companies with a higher proportion of fixed assets have lower leverage. This result was not confirmed for CSR companies. The hypothesis that CSR replaces the role of collateral cannot be confirmed. Available cash influences leverage negatively in both models, supporting the pecking-order theory. This result is much stronger for non-CSR companies compared to CSR ones. This study found fewer statistically significant differences between CSR and non-CSR companies regarding capital structure determinants than were expected.

**Keywords:** corporate social responsibility; leverage; capital structure; behavioral finance; non-financial companies

## 1. Introduction

More than six decades have passed since Modigliani and Miller (1958) introduced their progressive irrelevance theory of capital structure. It triggered new approaches and a plethora of empirical studies. No dominant version emerged from what became an often confusing comparative empirical inquiry.

Behavioral economics has led to a new way of looking at economics and finance, including capital structure. Indeed, several theories of capital structure make explicit behavioral assumptions a part of their fundamental structure. Non-financial, behavioral determinants of capital structure can partly plug the gaps left by research based solely on hard financial data. But the full complexity of the factors determining managerial decisions is still unknown (Bouzguenda 2018).

Concepts of corporate social responsibility (CSR), ethics, sustainability, and environmental, social, and corporate governance (ESG) have become an inseparable part of the discourse of modern business. They can help with brand building, and stakeholders expect to encounter them, especially when dealing with listed companies. Researchers have focused on the impacts of CSR/ESG on company value (e.g., Lu et al. 2021), cost of capital (e.g., Girerd-Potin et al. 2011; Benlemlih 2017; Cantino et al. 2017; Bae et al. 2019; Yeh et al. 2020) and leverage (e.g., Pijourlet 2013; Yang et al. 2016). Empirical studies often pay attention to CSR as a leverage determinant, alongside financial determinants

(e.g., Girerd-Potin et al. 2011), although Rahman and Alsayegh (2021) applied the opposite approach, as they considered CSR a dependent variable, while leverage was an independent variable.

But do capital structure determinants and their impact on leverage differ between CSR and non-CSR companies? This paper attempts to answer that question by analyzing how CSR can influence the relation between capital structure and its determinants, i.e., to identify the mediational effect of CSR on this relation through a comparison of capital structure models developed separately for CSR and non-CSR companies. This approach also represents a linkage with the study by Szymańska et al. (2015), although they focused on non-profit and for-profit companies. They found that tangibility has a negative influence on leverage in non-profit organizations, but a positive influence in for-profit companies. The capital structure of social purpose companies is also influenced by the sort of growth opportunities that we might expect to be more influential for for-profit companies.

## 2. Literature Review

"There is no universal theory of debt-equity choice, and no reason to expect one" (Myers 2001, p. 1). In fact, theoretical studies have explored a range of views on capital structure and its determinants, while empirical studies have revealed differing, indeed sometimes conflicting results. Here we focus on research on capital structure studies, with an emphasis on their behavioral aspects, alongside CSR studies (note: if it is not necessary and obvious from context, we do not distinguish between CSR and ESG).

### 2.1. Capital Structure Theories Review

To begin, Modigliani and Miller (1958) created the irrelevance theory, also known as the MM model, or the model of risk classes. With this theory there is no correlation between the value of a firm and its capital structure. The value of a firm depends on expected earnings before interest, and on risk class. In their subsequent theory, Modigliani and Miller (1963) allowed for the existence of tax. Then, the value of a firm is positively related to leverage, due to the interest tax shield. Firms are thus motivated to use interest-bearing debt, that is to use this shield, especially if the corporate income tax rate is relatively high.

However, there is an exception to this rule if the value of the tax base before deduction of interest is negative. That situation includes a naturally generated loss by a firm, optimization techniques (tax avoidance), as well as tax evasion. Ross (1985) and Drobetz and Fix (2003) also discuss "tax exhaustion", which means that not all shields and techniques to minimize tax can be used in parallel. For example, a firm can use the non-interest tax shield (depreciation and amortization) at such a level that there is no need to use the concurrent interest tax shield.

The trade-off theory (Kraus and Litzenberger 1973; Myers 1984) links the second MM (tax-including) model (Modigliani and Miller 1963) to bankruptcy theory (Stiglitz 1969; Scott 1977; Kim 1978). A firm needs to find a compromise between the advantages of an interest tax shield and the disadvantages of potential financial distress. This theory offers an interior solution for the debt-equity choice. It contrasts with the previous theories (which offer an irrelevant and a corner solution), but is in accordance with the so-called traditional U-curve approach. Financial distress costs can be divided into direct, such as law services, and indirect, for example investment restrictions, customers' and key employees' losses (Krištofík 2010), and lack of trust on the part of potential business partners and potential creditors. Reputational damage can follow previous owners and managers to new firms they have founded. As a result, firms with higher business risk, more volatile cash-flow, producing unique products (Krištofík 2002), doing business in a specific small B2B market, firms with greater investment opportunities whose realization can be endangered by bankruptcy (Režňáková et al. 2010), and firms that are more reliant on intangible assets and thus more sensitive to potential distress (Brealey and Myers 1992; Kráľovič and Vlachynský 2006) are all candidates for lower leverage. In addition, cash-flow volatility or business risk

itself are determinants of possible financial distress that can be amplified by high leverage. Other factors noted are linked to the consequences of financial distress.

Agency costs theory (Fama and Miller 1972; Jensen and Meckling 1976) is based on relations between different groups of stakeholders, whose interests may be mutually inconsistent. If a manager does not maximize shareholders' utility, it generates agency costs of equity. Moreover, there are costs related to controlling managers. But debt itself can stimulate managers to act effectively and control any impulses to spend resources on their personal consumption. However, debt can also generate agency costs; for example, its availability may lead managers to finance more risky investments than originally intended. These costs can be minimized via securities or covenants (Jensen 1986; Krištofík 2010). But if profitable investments are limited, and there is no shortage of ready cash to exploit them, then there is an argument that interest costs signal inefficient expenses.

These determinants of capital structure are considered in the pecking-order theory (Myers and Majluf 1984). Managers do not set the target leverage, but they use capital sources in a given order—firstly internal funds, secondly debt issue, and finally equity issue. Therefore, there is no debt issue if internal funds are adequate for all possible investments with positive NPV. In fact, that is a path of least resistance. We can find a connection to equity in the first and last positions in the mentioned order, and debt is in the middle. Consequently, the determinants of capital structure, such as growth opportunities, dividends, cash, profitability, depreciation, and amortization—which are all in some way connected to the mentioned adequacy of internal funds—are not determinants of leverage, but of external sources. As share dilution is usually unwelcome to the original owners, thus it is the least probable outcome, we can also consider them as determinants of leverage.

The life cycle theory (Weston and Brigham 1981; Chittenden et al. 1996) indicates that capital structure can be affected by age, size, access to capital markets, and in accordance with previous theories, profitability, plus the collateral value of assets. Smaller and younger firms are usually financed by their original owners, and later by retained earnings. Because they must first obtain the trust of their creditors, their access to loans may be constrained. Alternatively, traditional creditors may supply loans if there is adequate collateral or covenants. Modern sources include venture capital and business angels. IPO and bond issues are impossible until firms qualify for access to capital markets. As a result, small and young firms risk only being able to achieve suboptimal investment levels.

On the other hand, market-timing theory focuses on listed companies, which usually do not face financial constraints. The theory supposes that listed companies issue equity when their stock price is above average, or when the market price of equity exceeds its book value. When the reverse is true, they make redemptions. So, the capital structure of a company is a result of previous attempts at market-timing. (Baker and Wurgler 2002). We can also apply this theory to debt issue. A company is motivated to issue more debt when interest rates are low, and vice versa.

To conclude, Table 1 summarizes the determinants of capital structure according to the various noted theories.

**Table 1.** Comparison of capital structure theories according to its determinants.

| Determinant/Theory | MM Tax-Including Model | Trade-Off Theory | Agency Costs Theory | Pecking-Order Theory | Life Cycle Theory | Market-Timing Theory |
|---|---|---|---|---|---|---|
| Effective corporate income and dividend [1] tax rate | + | + | | | | |
| Effective interest income tax rate [1] | - | - | | | | |
| Depreciation & amortization | - [2] | - [2] | | - | | |
| Tax optimization | - [2] | - [2] | | | | |
| Profitability | + | + [3] | | - | - | |
| Business risk | | - | | | | |
| Bankruptcy costs | | - | | | | |
| Cash | | | + | - | | |
| Growth opportunities | | - | | + | | |
| Rating | | | + | | + | |
| Dividends | | | | + | | |
| Asset tangibility (collateral) | | + | + | | + | |
| Size and age | | | | | + | |
| Market to book ratio [4] | | | | | | - |
| Interest rate | | | | | | - |

Note: The table's symbols indicate the direction of the determinants' impact on leverage. [1] Dividends tax rate (paid by shareholders) and interest income tax rate (paid by creditors) are considered as capital structure determinants according to Miller's tax index. [2] If a firm does not face with a tax-shield exhaustion, interest and non-interest tax shield (another optimization technique, respectively) can be used in parallel. Then, the effect of D&A on leverage is not necessarily negative. A positive correlation between leverage and D&A, if it occurs, can be also caused by a positive correlation between leverage and tangibility (D&A and tangibility relations are obvious but depend on used measures). [3] The positive influence of profitability is consistent with the use of an interest tax shield i.e., with the first part of the trade-off theory (consistent with MM tax-including model). Regarding the second (bankruptcy-oriented) part of the trade-off theory, we might assume that the more profitable the firm, the more easily it can solve financial problems. But the greater the instability of cash flows the weaker this link. In addition, higher expected profits mean higher indirect costs of a financial distress, as expected profits will never be achieved if a firm goes bankrupt. As a result, the direction of effect of profitability on leverage in the case of the trade-off theory can be questionable. [4] The market-to-book ratio is definitely the determinant of the market-timing theory. However, it has also been used as a measure of growth opportunities (another determinant) in some studies of other theories.

## 2.2. CSR-Oriented Studies in the Context of Capital Strcuture Theories as a Basis for Hypotheses Development

If CSR reduces information asymmetry and agency costs, that can raise investor trust and so reduce the influence of financial constraints (Cheng et al. 2014; Li et al. 2021). Therefore, we can consider CSR as a capital structure determinant connected to both the agency costs, and to the life cycle theories. For example, CSR activities can serve as a substitute for collateral in the process of building up trust with creditors. However, we should stress that CSR may have a positive influence on both potential shareholders, and potential creditors. So, its net effect on leverage may be unclear. We will show that CSR can be an important issue that should also be addressed by other capital structure theories. Therefore, we expect a statistically significant difference in the average/median leverage between CSR and other companies.

A clearer effect of CSR on leverage can be considered with reference to the moralization function of debt (related to the agency costs theory). We can expect that CSR companies' managers do not need debt to be moralized, and that they would not routinely use available cash ineffectively for personal managerial consumption. The opposite connection between CSR and agency costs theory can be observed if CSR investments are made for the private benefit of managers, e.g., if they want to be considered as good citizens (Barnea and Rubin

2006; Yeh et al. 2020). Their enhanced social status, caused by the CSR investments, can be considered as a "CSR alternative" to traditional managerial ineffective consumption related to the agency costs theory (and available cash as a leverage determinant), if these private benefits for managers are the aim of the CSR investments.

Socially responsible companies are expected to pay adequate taxes (Abdelfattah and Aboud 2020). So, CSR disclosures can be contrasted with the use of tax shields, and thus in part contrasted with the MM tax-including model, as well as with the trade-off theory; because of reputational risk. On the contrary, CSR activities can also serve as an "excuse" for tax avoidance activities (Lin et al. 2017; Abdelfattah and Aboud 2020). Thus, they can be consistent with the use of tax shields noted in previously mentioned theories.

For completeness, we should note an alternative viewpoint. Tax avoidance may be used instead of tax evasion to achieve the same objectives. Consequently, tax avoidance can be a positive policy, suitable for CSR companies, because it naturally reduces tax evasion. Moreover, legal tax minimization can be considered a shareholder-friendly policy. Therefore, we cannot produce an unambiguous prediction for the net effect of CSR influence on the use of tax shields.

With reference to riskiness, CSR companies can be perceived as less risky than others (Verwijmeren and Derwall 2010; Yeh et al. 2020). That provides a link between CSR and the trade-off, and bankruptcy theories. However, the opposite view includes a consideration of indirect bankruptcy costs, e.g., risk of staff redundancies. Therefore, CSR companies which pay attention to employee well-being should reduce their leverage, and thus reduce bankruptcy risk.

In addition, as stock prices can be higher after CSR disclosures—because of demand for stocks by CSR-focusing individuals or institutional investors, because of a risk mitigation effect (Lu et al. 2021), and because of an efficiency enhancement effect—the link between CSR and the market-timing theory can also be considered.

Finally, the pecking-order theory can be used in reverse order—internal funds, equity issue, debt issue both for CSR companies (Pijourlet 2013; Benlemlih 2017), and companies run by overconfident mangers (Bukalska 2019). We can also claim that a lack of available cash should lead to equity issue instead of debt issue for CSR companies. If we also want to compare companies run by overconfident mangers to CSR companies, we should look at leverage adjustment speed. Bukalska (2019) discovered that overconfident managers adhere to target leverage, while CSR companies have slow adjustment speeds (Yang et al. 2016; Akhtar et al. 2016).

In accordance with these conclusions of the literature review, we have developed the following set of hypotheses.

**Hypothesis 1 (H1).** *Leverage of CSR companies differs from leverage of other companies.*

**Hypothesis 2 (H2).** *Effective corporate income tax rates and depreciation and amortization influence leverage significantly differently in CSR companies than in other companies.*

**Hypothesis 3 (H3).** *Asset tangibility positively influences the leverage of non-CSR companies. This influence is significantly less for CSR companies.*

**Hypothesis 4 (H4).** *Available cash influences leverage significantly differently for CSR companies compared to others.*

To sum up, our hypotheses anticipate that the traditional determinants of capital structure, based on the selected theories, will apply differently to CSR and non-CSR companies (in general, and regarding to tax shields, collateral, available cash, etc.). This approach also represents a linkage with the study by Szymańska et al. (2015), although they focused on non-profit and for-profit companies.

The synthesis of traditional and behavioral determinants along with their context is presented in Table 2, as well. A deeper view on CSR is applied there, as we distinguish

between several dimensions of CSR (community, employees, governance). The advantage of this approach is that CSR dimensions can be separately linked to capital structure theories along with their expected (positive/negative) effect on leverage. However, practical applications for further research can suffer from multicollinearity of CSR dimensions, which is problematic especially due to the fact that we anticipate different directions of impact on leverage for each CSR dimension.

**Table 2.** Addition of traditional determinants of capital structure by behavioral ones.

| Traditional/Financial Determinant (and Its Effect on Leverage) | Theory | Context | Behavioral Determinant | Effect of Behavioral Determinant on Leverage |
|---|---|---|---|---|
| Collateral and covenants (+) | Agency costs, Life cycle | Motivating investors by mitigating the risk of their investment | CSR governance and overall | + (CSR replaces collateral) |
| Business risk (-) | Trade-off | Less risky company can afford to have higher leverage | CSR governance and overall | + (CSR = higher safety) |
| Indirect bankruptcy costs: (lost) growth opportunities (-) | Trade-off | Risk of non-implementation of business or CSR projects because of a financial distress | CSR community | - (aim is to mitigate bankruptcy risk) |
| Indirect bankruptcy costs: loss of key employees (-) | Trade-off | Risk of staff redundancies because of a financial distress | CSR employees | - (aim is to mitigate bankruptcy risk) |
| Available cash (+) | Agency costs | Motivating managers to reduce inefficient consumption (through debt or better governance) CSR alternative to inefficient managerial consumption = community projects implemented in order to be perceived as a "good citizen" | CSR governance CSR community | - (CSR replaces moralization function of debt) + (moralization function of debt is needed to avoid inefficient consumption) |
| Interest tax shield (+) | MM tax-including model, Trade-off | Less motivation for CSR companies to use the shield, as they are expected to pay "adequate" taxes Increased use of the shield while "justifying" optimization by CSR activities | CSR overall CSR community | - (interest tax shield is not needed) + (interest tax shield is used) |

## 3. Materials and Methods

### 3.1. Variables

The variables in this study can be divided into three groups. First, the dependent variable is leverage. Second, the independent variables are the effective corporate (income) tax rate, depreciation and amortization, asset tangibility, and available cash. Third, the moderating variable, as well as our non-financial behavioral item, is CSR.

We will use two models. Both are linear regression models of leverage and the independent variables, thus, it is originally the same model, but used on two subsamples. Model 1 uses CSR companies' data, while Model 2 is constructed from other companies' data.

Leverage is calculated as the ratio of total liabilities to equity (LIABE); book values are used. Such a ratio gives an overall view across all financial sources. Market values are not used because of possible CSR influence on stock price, and consequently on the market value of equity (observed e.g., by Iqbal et al. 2012; Lu et al. 2021). We cannot be sure whether this influence would overwhelm some of the effects we want to study.

Effective corporate (income) tax rate (TAX) is calculated as the ratio of paid income tax and EBT. Alternatively, if that is not possible (negative values), we use the statutory nominal corporate income tax rate. This determinant of capital structure has been empirically confirmed in several studies (e.g., Faccio and Xu 2015; Orjinta and Agubata 2017; Rehman et al. 2019).

Depreciation and amortization (DA or D&A) with a negative direction of influence on leverage has been observed, e.g., by Krištofík (2002); Režňáková et al. (2010); M'ng et al. (2017). The ratio of D&A to total assets is used as the variable.

For asset tangibility (TAN), we prefer the often-used ratio of fixed to total assets. The positive influence of asset tangibility on leverage is confirmed in many empirical studies (e.g., Krištofík 2002; Drobetz and Fix 2003; Chen and Zhao 2006; Akdal 2010; Bhaird and Lucey 2010; Režňáková et al. 2010; Faccio and Xu 2015; M'ng et al. 2017).

Available cash (CASH) is calculated as liquidity because we want to consider the necessity of the settlement of current debt. To be more specific, quick ratio is used, because it includes the most liquid current assets that can be used for this settlement in the short run. Empirically, a positive correlation between leverage and cash was observed by Mahadwartha and Ismiyanti (2008), and a negative one by Mouline and Sadok (2021).

CSR is our moderating variable. Therefore, it is measured as a dummy variable (0 for non-CSR; 1 for CSR). CSR is used as a dummy variable, e.g., by Girerd-Potin et al. (2011) when they distinguish between top CSR companies, and companies with the lowest CSR scores.

To sum up, our model can be written as follows (standard legend for linear regression is used).

$$\text{LIABE}_i = \beta_0 + \beta_1 \, \text{TAX}_i + \beta_2 \, \text{DA}_i + \beta_3 \, \text{TAN}_i + \beta_4 \, \text{CASH}_i + \varepsilon_i, \tag{1}$$

For Model 2 (non-CSR companies), we can anticipate such directions of the relations between leverage and capital structure determinants as they are presented in Table 1, while for Model 1 (CSR-companies) this cannot be specified unequivocally, but in the context of the hypotheses.

*3.2. Methods for the Hypotheses Testing*

**Hypothesis 1 (H1).** *Leverage of CSR companies differs from leverage of other companies.*

The hypothesis H1 will be confirmed if Student *t*-test (for independent samples) and Wilcoxon rank sum test, respectively, records a statistically significant difference between the two samples.

**Hypothesis 2 (H2).** *Effective corporate income tax rates and depreciation & amortization influence leverage significantly differently in CSR companies than in other companies.*

The hypothesis H2 will be confirmed, firstly, if the beta-coefficient before the variable effective corporate income tax rate in a leverage model of CSR companies (Model 1) is significantly different from those in a leverage model of other—non-CSR—companies (Model 2), and secondly, if the beta-coefficient before the variable depreciation and amortization differs significantly between Model 1 and Model 2.

**Hypothesis 3 (H3).** *Asset tangibility positively influences the leverage of non-CSR companies. This influence is significantly less for CSR companies.*

The hypothesis H3 will be confirmed if the beta-coefficient before asset tangibility is significantly lower in Model 1 than in Model 2.

**Hypothesis 4 (H4).** *Available cash influences leverage significantly differently for CSR companies compared to others.*

The hypothesis H4 will be confirmed if the beta-coefficient before available cash in Model 1 is significantly different from that in Model 2. We cannot expect a certain direction of CSR influence.

Regarding methods, in addition to those mentioned, we use also a linear regression model and Cumming's (2009) method for evaluating differences between beta-coefficients. This method compares the betas' 95% confidence intervals. If they do not overlap by more than 50%, the betas can be taken as significantly different at the 5% level.

In more detail, a proportion overlap (for two independent groups) is measured, as follows. $\beta m_1$ and $\beta m_2$ are means of confidence intervals (i.e., standardized regression coefficients for the same independent variable in the Model 1 and Model 2), $u_1$, $u_2$ are upper limits and $k_1$, $k_2$ are lower limits of the standardized confidence intervals. The margins of error are $w_1 = (u_1 - k_1)/2$; $w_2 = (u_2 - k_2)/2$. In general, margin of error can also be calculated as $w = u - \beta$ or $\beta - k$. The average margin of error $w = (w_1 + w_2)/2$. If $\beta m_1 > \beta m_2$, then overlap is calculated as $O = u_2 - k_1$. Proportion overlap $PO = O/w$. PO must be compared to the threshold 50% (i.e., 0.5) (Cumming and Finch 2005).

These methods allow us to study not only the effect of CSR on capital structure, but also the mediational effect of CSR on the relations between capital structure and its determinants. We prefer these methods as our aim is to show whether there is a difference between CSR and non-CSR companies regarding these relations.

*3.3. Data*

The sample consists of non-financial large-cap companies from the STOXX Europe 600 index (non-financial companies with market capitalization over 10 billion USD as at March 2020). In accordance with the mentioned methodology, two sub-samples have been created—CSR and non-CSR. The selection is based on a wide range of CSR lists. Diez-Cañamero et al. (2020) note plenty of suitable indexes, rankings, and ratings. We have adopted some of them. Together with our own selection, the CSR lists used are as follows—World's Most Ethical Companies; CDP Climate, Water, and Forests A lists; CSR Europe's corporate members; Top 100 Most Reputable Companies in the World—Global RepTrak; The Sustainability Yearbook (medal holders only); Vigeo Eiris' World 120 Index; Euronext Vigeo EU 120 Index; EURO STOXX Sustainability 40 Index; CSR Hub's rating (a component of the 100 best non-financial companies in the European region); Horizons Global Sustainability Leaders Index; and MSCI ESG Rating (triple-A holders only).

As there are many lists, rankings, and ratings, any selection between them will be at least partly subjective. In this context, Rahman and Alsayegh (2021) suggest that some authority should try to create a consensus in CSR reporting. Arribas et al. (2019) go as far as claiming that the well-known Dow Jones Sustainability Index includes some arguably irresponsible companies. Such claims make justification of the sample even more difficult.

Every European non-financial large cap company has been classified to the CSR subsample provided it appears in any of the selected indices and major lists noted above. Small- and mid-caps are not studied because their "chance" to be a component of CSR top lists and indexes are lower, thus the described methodology would be unsuitable for them. Moreover, a positive effect of size on CSR has been observed, for example by Rahman and Alsayegh (2021), and size is also a determinant of capital structure. Because we only include large caps, our data are less company-size sensitive. In addition, we must stress that CDP A lists only focus on the environment: other CSR dimensions are not considered by CDP. However, Cheng et al. (2014), in their study of access to finance, observe similar

results for the cases of the social and environmental dimensions of CSR. Thus, we do not consider CDP ranking as unsuitable.

For the sub-samples, after the exclusions, there remain 104 CSR, and 59 non-CSR companies. One observation has been excluded because of a minus value for equity, and one outlier has been excluded (see Appendix A). Regarding time, the financial statements date from the last pre-crisis year 2019, and were obtained from Thomson Reuters Eikon (Thomson Reuters 2020), and Investing—Stocks—Financials (2021), while the CSR lists used date from 2020.

### 3.4. Econometric Tests of the Models

Both models have been tested to meet econometric conditions for LRM (linear regression model). The results are presented in Table 3. In accordance with the results of the Ramsey reset test, we have used a log-log model for CSR companies (Model 1). Log-lin, lin-log, as well as lin-lin models for CSR companies did not suggest a correct model specification. Our hypotheses and methodology focus on a comparison of beta coefficients between Model 1 and Model 2. To make beta coefficients comparable, we have also used a log-log model for non-CSR companies (Model 2). The Ramsey reset test confirms the correctness of Model 2's specification, but only at the 1% level, not at the 5% level (see Table 3). The White test shows homoskedasticity for Model 1, but the *p*-value is just slightly above 5%. Therefore, we applied the Breusch–Pagan–Godfrey test, which confirmed homoskedasticity at all standard significance levels. The remaining conditions for LRM are fulfilled without discrepancies. To sum up, tests show normal distribution of residuals, homoskedasticity, and no multicollinearity (see Table 3).

**Table 3.** Results of econometric tests.

| Test/Statistics | Null Hypothesis | *p*-Value or Value of VIF (Model 1, CSR) | *p*-Value or Value of VIF (Model 2, Non-CSR) |
|---|---|---|---|
| Jarque–Bera of residuals | Normal distribution | 0.2038 | 0.9459 |
| White | Homoskedasticity | 0.0562 | 0.1170 |
| Breusch–Pagan–Godfrey | Homoskedasticity | 0.1140 | 0.2759 |
| Ramsey reset test | Correct specification | 0.4532 | 0.0343 |
| VIF for TAX | | 1.0517 | 1.0375 |
| VIF for DA | | 1.0157 | 1.4078 |
| VIF for TAN | | 1.2437 | 2.3777 |
| VIF for CASH | | 1.3010 | 2.4256 |

As we mentioned, the log-log models are applied. Therefore, the original Equation (1) is finally adjusted, as follows.

$$\log(\text{LIABE}_i) = \beta_0 + \beta_1 \log(\text{TAX}_i) + \beta_2 \log(\text{DA}_i) + \beta_3 \log(\text{TAN}_i) + \beta_4 \log(\text{CASH}_i) + \varepsilon_i, \quad (2)$$

## 4. Results and Discussion

First of all, we should look at leverage and hypothesis H1. The leverage variable is not normally distributed, with the *p*-value almost zero in both the Jarque–Bera and Shapiro–Wilk tests. Therefore, we used the non-parametric Wilcoxon rank sum test (aka Mann–Whitney test). This shows a statistically significant difference (*p*-value 0.0099) between the median values of leverage of CSR and non-CSR companies. We observe that CSR companies are more leveraged (see Figure 1; for more details see Appendix A). The hypothesis H1 is confirmed. Moreover, not just the median and average values of leverage differ between non-CSR and CSR companies. Minimum, first quartile, last quartile, and maximum values are all clearly higher in the case of CSR companies. Such results are in accordance with, e.g., Yang et al. (2016) who also observed a positive effect of CSR on leverage.

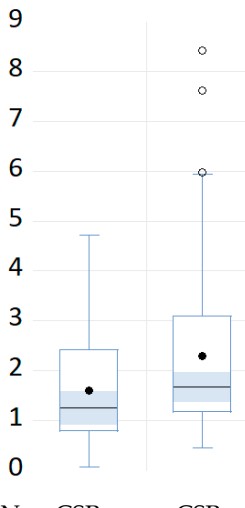

**Figure 1.** Leverage (LIABE) of non-CSR and CSR companies. One extra, wide outlier (in addition to the outlier that we mention in Section 3 Data) has been excluded just for the analysis of the hypothesis H1. After logarithmic transformation (actually log-log LRMs are used for analysis of the hypotheses H2, H3, H4) this observation is not an outlier. In fact, the extra outlier would only magnify the measured difference related to hypothesis H1.

On the contrary, in studies reviewed by Cantino et al. (2017), lower costs of equity for CSR companies are observed in comparison to non-CSR companies. Moreover, Pijourlet (2013), and Benlemlih (2017) claim that the reverse pecking-order theory is suitable for CSR companies. Both of these findings should lead to equity preference. Pijourlet (2013), and Benlemlih (2017) both claim that CSR positively affects not only equity preference, but also the amount of equity issued. We definitely cannot confirm these findings for our sample. The idea that CSR companies are motivated to lower their leverage to mitigate a bankruptcy risk and its indirect costs (such as staff redundancies) is not supported, as well. But we are able to observe a similarity between CSR and non-CSR companies—a right-sided distribution of leverage.

Second, we can take a closer look at relations between leverage and its selected determinants. The regression models (see Table 4) show that only available cash is statistically significant at all levels in both models. According to a negative direction of its beta coefficients, the pecking-order theory is partly confirmed for both CSR and non-CSR companies. We can say that companies that have sufficient available cash do not need to issue debt. Asset tangibility is statistically significant at the 10% level only in Model 2. But the direction of its beta-coefficient is not consistent with the predictions of any of the theories. Other capital structure determinants are insignificant. However, we retain all of the variables in both models to focus on a comparison of the betas' confidence intervals.

**Table 4.** Model 1 and Model 2.

| Variable | Model 1 | | Model 2 | |
|---|---|---|---|---|
| | Coefficient | Standardized Coefficient | Coefficient | Standardized Coefficient |
| Intercept | 0.8566 * | - | −0.0719 | - |
| TAX | 0.0672 | 0.0548 | 0.1225 | 0.0768 |
| DA | 0.0427 | 0.0306 | −0.1013 | −0.0957 |
| TAN | 0.1805 | 0.0643 | −0.4346 * | −0.3080 |
| CASH | −0.5909 *** | −0.3758 | −0.9625 *** | −0.7984 |

Note: ***, * statistical significance at 1%, and 10% level, respectively.

The confidence intervals are presented in Figure 2. For the effective tax rate, there are remarkably similar results for CSR and non-CSR companies. Even though a difference in the case of D&A is also statistically insignificant, we can see a different direction of standardized coefficients (means of intervals). While CSR companies with higher D&A also have higher leverage, for non-CSR companies the opposite is true. There are three possible explanations. First, non-CSR companies do not need an interest tax shield if they have a non-interest tax shield. Consequently, they act in accordance with the MM tax-including model, with the trade-off theory and tax avoidance (if we consider the shields as substitutes, not complements). Such a result shows that tax avoidance activities can be more important for non-CSR, than for CSR companies. That is consistent with a minimization of reputation risk by CSR companies (Abdelfattah and Aboud 2020). However, D&A also occurs as a determinant of capital structure in the pecking-order theory. Thus, a second possible explanation is simply that this confirms the pecking-order theory (for non-CSR companies). However, D&A is statistically insignificant in both Model 1 and in Model 2 that does not support the previous interpretations. Confidence intervals differ, but not significantly. Therefore, the third explanation is that there is no significant difference between CSR and non-CSR companies with respect to D&A. Since there is also no difference between the groups of companies with respect to effective corporate tax rate, the hypothesis H2 is not confirmed. The insignificant differences in these cases also indicate that policy makers should not perceive CSR companies and non-CSR companies differently when considering tax avoidance.

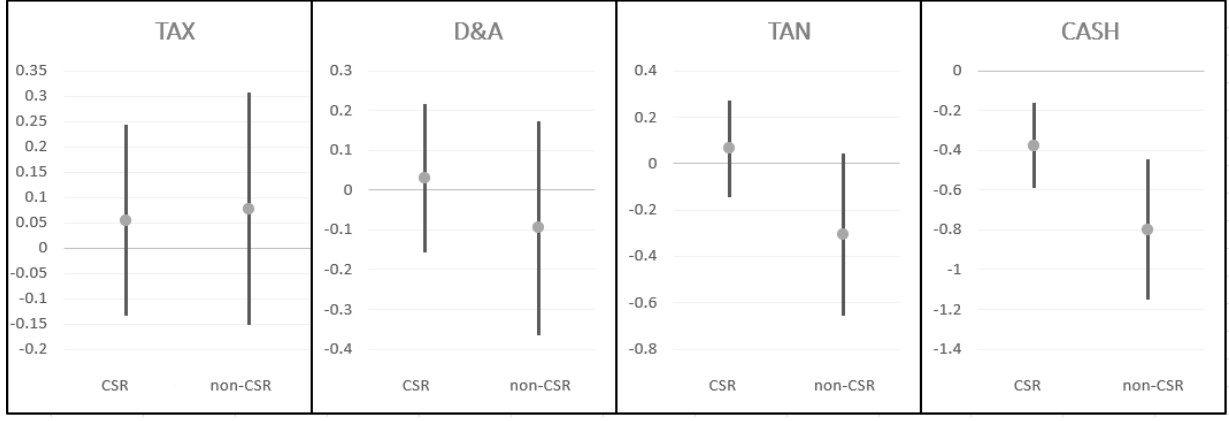

**Figure 2.** Confidence intervals of standardized beta-coefficients.

The confidence intervals of asset tangibility differ from each other (Figure 2). This difference is very close to being statistically significant at the 5% level, according to Cumming's (2009) method. So, for this reason, we view the difference as worth noting. Surprisingly, the interval for non-CSR companies is lower, than those for CSR companies. The opposite result would be expected. CSR companies should achieve investors' trust without needing collateral, while non-CSR companies should be in a worse position, because they may need collateral. In fact, the beta-coefficient before asset tangibility is positive, but not statistically significant for CSR companies. Thus, their need of collateral is small or non-existent. The negative value for non-CSR companies is not explained by the theories we have used. However, we should consider a fact of a calculation. We have used the ratio of fixed assets to total assets for this variable. Therefore, the negative value can be interpreted in following way. Non-CSR companies, with relatively higher fixed assets, have lower leverage. We should reconsider a calculation. Leverage is calculated as the ratio of total liabilities (both long-term and short term) to equity. Since short-term financial sources increase this ratio, the negative relation between leverage and the ratio of fixed to total assets may be interpreted as an application of a "golden rule". Whereas non-CSR companies are known to be riskier than the CSR ones (Verwijmeren and Derwall 2010; Yeh et al. 2020), a fulfilment of the rule is especially suitable for them. In contrast, less

risky CSR companies can afford more aggressive financing. This explanation definitely suggests further research. We did not distinguish between long-term and short-term debt in calculating leverage. Consequently, our explanation is only a conjecture, not yet based on observation. However, what can be stated is that the assumption that CSR replaces collateral cannot be confirmed.

Some parallel of our results with Szymańska et al. (2015) should also be highlighted, even though they studied differences between for-profit and non-profit organizations. Like us, they also observed between-group differences in the relations between leverage and asset tangibility. They showed that tangibility has a negative influence on leverage in the case of non-profit organizations, and a positive influence in the case of for-profit ones. If we want to liken CSR companies to non-profit organizations, and non-CSR companies to for-profit organizations (although this is far from reality), Szymańska et al. (2015)'s results are the opposite of ours.

Given the hypotheses we developed, hypothesis H3 would be confirmed if the beta-coefficients in Model 1 were significantly lower than those in Model 2. The reality is that the opposite is true, so H3 is not confirmed.

The final variable is available cash. Figure 2 shows a difference that is statistically significant at the 5% level, according to Cumming's (2009) method. Thus, the hypothesis H4 is confirmed. Both CSR and non-CSR companies with higher amounts of available cash have lower leverage. That is in accordance with the pecking-order theory. For CSR companies, this relation is much weaker than for the non-CSR companies. In itself that would show a partial applicability of the reverse pecking-order theory, more strongly for CSR companies (Pijourlet 2013; Benlemlih 2017) than for non-CSR ones (however, this interpretation is not supported by our results presented in Figure 1). In addition, it would show an application of the moralization function of debt that can minimize ineffective CSR overinvestments (Barnea and Rubin 2006; Goss and Roberts 2011; Yeh et al. 2020). But as the sign of the beta coefficient is also negative for CSR companies, we do not have an unequivocal confirmation of these claims. Given the observed significant differences of beta-coefficients for this variable, it will require further research to explain all of the differences in detail.

## 5. Conclusions

CSR can be related to capital structure theories, and their applicability. However, theoretically we cannot identify definite CSR influences on the relations between capital structure and its determinants; although applying CSR dimensions is useful for an identification of deeper theoretical relations. In relation to the MM tax-including model and the trade-off theory, tax shields influence capital structure. CSR companies should not use "too much" tax avoidance, because of reputational risk (Abdelfattah and Aboud 2020). On the other hand, companies that pay low taxes can make CSR disclosures to excuse their tax avoidance (Lin et al. 2017; Abdelfattah and Aboud 2020). The trade-off theory also assumes that companies with higher business risk should have lower leverage. CSR companies are known as less risky (Verwijmeren and Derwall 2010; Yeh et al. 2020); therefore, their leverage can be higher. In relation to agency costs theory, CSR in itself lowers information asymmetry and agency costs (Cheng et al. 2014; Li et al. 2021). However, if managers of CSR companies make inefficient CSR overinvestments merely to attract attention to themselves, this will adversely affect agency costs (Barnea and Rubin 2006; Yeh et al. 2020).

The life cycle theory and agency costs theory suggest that the existence of collateral can raise investor trust. Thus, CSR companies, that should naturally be more trusted than non-CSR companies, have less need of collateral. In a sense, CSR disclosures replace collateral. CSR companies also have lower equity costs than non-CSR companies (Cantino et al. 2017). However, they can have the same debt costs (Yeh et al. 2020). As a result of these, and of other nonmentioned linkages between CSR and capital structure, we cannot form definite conclusions in this area.

We have focused on the following capital structure determinants—effective corporate (income) tax rate, depreciation and amortization, asset tangibility, and available cash. They are our independent variables. Leverage is our dependent variable. Two linear regression models with the same dependent and independent variables have been developed for CSR and for non-CSR companies. Thus, CSR serves as the moderating variable. The aim of the paper has been to identify the impact of CSR on leverage, and on the relations between capital structure and its determinants.

We have observed significant inter-group differences in leverage. That shows strong debt preference by CSR companies. However, we have identified almost no difference between CSR and non-CSR companies for the relation between leverage and the effective corporate (income) tax rate. Only a statistically insignificant difference was observed for the relation between leverage and depreciation and amortization. However, we should stress that the coefficient for depreciation and amortization has a different sign for CSR-companies compared to non-CSR companies. But the difference is not significant. With reference to both interest and non-interest tax shields, we can conclude that policy makers should not perceive CSR companies and non-CSR companies differently when considering tax avoidance.

Somewhat stronger is the nearly statistically significant difference between CSR and non-CSR companies regarding the influence of asset tangibility on leverage. So, there is some evidence that non-CSR companies, with a higher proportion of fixed assets, prefer equity financing, while CSR ones do not. The cash variable is statistically significant, and negative in both models. But its influence on leverage is significantly much stronger for non-CSR companies. As the sign on the coefficient is the same in both models, we cannot make a more conclusive finding. Future research, especially regarding this issue, is surely needed.

It is also needed with reference to the limitations of our research, especially in terms of data and methodology. One of the most important limitations is a selection of CSR rankings, ratings, and lists. Furthermore, CSR as a dummy variable allowed us to measure its overall moderating role in the examined relations, but CSR scores reflecting its dimensions (such as community, employees, environment, and governance) can also be applied in a capital structure model along with traditional determinants, as we showed in Table 2. Our contribution to theoretical knowledge rests especially in the way we summarized the linkages between well-known capital structure theories, traditional determinants of capital structure, respectively, and the dimensions of CSR. In theory, some CSR dimensions can have a positive, others a negative, impact on leverage. Furthermore, the impact differs between capital structure theories. As a result, relations between leverage and its determinants do not necessarily have to differ significantly when overall CSR impact is examined.

**Author Contributions:** Conceptualization, P.K. and J.M.; methodology J.M.; formal analysis, J.M.; data curation, P.K.; writing—original draft preparation, P.K. and J.M.; writing—review and editing, H.M.; visualization, P.K.; supervision, P.K.; project administration, H.M. and P.K.; funding acquisition, H.M. All authors have read and agreed to the published version of the manuscript.

**Funding:** This research was funded by the Scientific Grant Agency of the Slovak Republic within the project VEGA no. 1/0579/21 "Research on Determinants and Paradigms of Financial Management in the context of the COVID-19 Pandemic".

**Institutional Review Board Statement:** Not applicable.

**Informed Consent Statement:** Not applicable.

**Data Availability Statement:** Financial data of the examined companies are available online: https://www.investing.com/equities/ (accessed on 19 April 2021), or https://eikon.thomsonreuters.com/index.html (accessed on 1 March 2020). CSR ratings/rankings/lists which were applied, are available online: https://worldsmostethicalcompanies.com/past-honorees/ (accessed on 25 February 2021), https://www.cdp.net/en/companies/companies-scores (accessed on 15 March 2021), https://www.

csreurope.org/our-network-1 (accessed on 5 February 2021), https://www.reptrak.com/rankings/ 2020/ (accessed on 21 March 2021), https://www.spglobal.com/esg/csa/yearbook/files/482663 _RobecoSAM-Year-Book_Final_med.pdf (accessed on 3 March 2021), https://live.euronext.com/ popout-page/getIndexComposition/QS0011250873-XAMS (accessed on 10 March 2021), https:// www.stoxx.com/index-details?symbol=sube (accessed on 12 March 2021), https://www.csrhub.com/ search/country/region/Europe/and/overall/greater_than/65/but_not/industry/group/Finance-and-Real-Estate?page=1 (accessed on 1 March 2021), https://horizonsetfs.com/ETF/ethi/#holdings (accessed on 2 April 2021), https://www.msci.com/our-solutions/esg-investing/esg-ratings-climate-search-tool (accessed on 14 March 2021). Note: Composition of lists (especially indexes) is subject to change.

**Conflicts of Interest:** The authors declare no conflict of interest. The funders had no role in the design of the study; in the collection, analyses, or interpretation of data; in the writing of the manuscript, or in the decision to publish the results.

## Appendix A

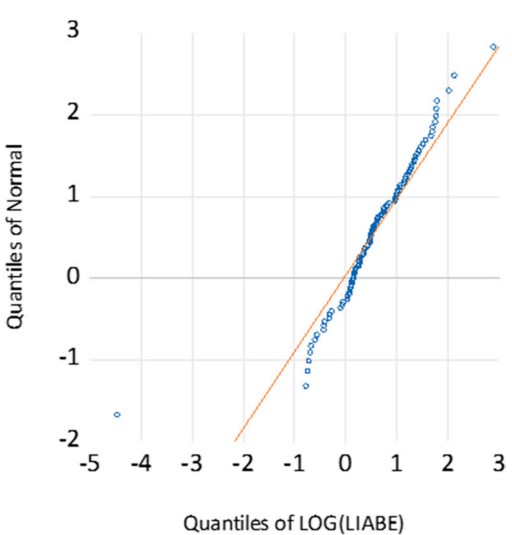

**Figure A1.** The evidence of outlier—QQ plot of LIABE (after logarithmic transformation) before an exclusion of outlier (data of CSR companies).

**Table A1.** Leverage of CSR and non-CSR companies.

| CSR | Mean | Median | Standard Deviation | Observations |
|---|---|---|---|---|
| non-CSR | 1.6142 | 1.2647 | 0.9994 | 59 |
| CSR | 2.2936 | 1.6830 | 1.6330 | 103 |
| all | 2.0462 | 1.6344 | 1.4686 | 162 |

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
