# Peer review of "Capital Structure and Its Determinants—A Comparison of European Top-Rated CSR and Other Companies"

_jrfm, doi:10.3390/jrfm15080325_

Round 1

Reviewer 1 Report

The paper Capital Structure and its Determinants – A Comparison of European Top-Rated CSR and other Companies is written and presented with details in the research steps and results.

 The first part of the paper presents the theories on the structure of capital and its determinants and then the results of empirical research identified in published articles.

There are research hypotheses constructed and empirically tested in this paper.

 Some minor points are required to improve or clarify:

1.      I think it would be good to detail the method of Cummings (2009).

2.      Why not use dummy variables in the regression model that allows, more easily, to test the difference between the two categories of companies.

3.       If the log-lin, lin-log, ... models are estimated, the estimated regression model is not the one presented in equation 1.

Author Response

Point 1: I think it would be good to detail the method of Cummings (2009).

Response: Comment accepted, the Cumming´s method was described in the revised version in more details incl. why we used just this method. 

Point 2: Why not use dummy variables in the regression model that allows, more easily, to test the difference between the two categories of companies.

Response: Due to usage of above mentioned Cumming´s method it wouldn´t be appropriate to apply dummy variables in the regression model. 

Point 3: If the log-lin, lin-log, ... models are estimated, the estimated regression model is not the one presented in equation 1.

Response: Comment accepted. The equation for log-log model was added.

Reviewer 2 Report

Topic

The topic is not exciting to the reader. Not very innovative.

Introduction

It has to be clear identification of the gap in the literature.

Refer to the studies that justify the need for the study.

What is new about the paper compared to the existing literature?

Literature review

It was supposed to do a literature review that included the topic under study without the need to include a point like point 2.2 (2.2. Review of empirical studies focusing on capital structure and CSR). It gives an idea of little art in the literature review.

Methodology

Research hypotheses should be included in the literature review. This is because the hypotheses derive from the literature review.

The context in which the study was conducted can be described.

The method followed can be more explanatory.

Results

It does not make sense to put together the presentation of results and the discussion of results. They are two distinct sections.

Figure 1 is not appropriate in the discussion of results.

Conclusions

The contributions of the study to the literature and to practitioners should be emphasized.

Author Response

Point 1: The topic is not exciting to the reader. Not very innovative.

Response: In our opinion, the link between capital structure and CSR aspects is currently very attractive to readers. If necessary, the title of the paper can be changed.

Point 2: It has to be clear identification of the gap in the literature. Refer to the studies that justify the need for the study. What is new about the paper compared to the existing literature?

Response: Comment accepted. References and citations pointing out the differences between our study and others are added in the introductory part. In this part also the contribution of our study to fill the gap in the literature can be found.

Point 3: It was supposed to do a literature review that included the topic under study without the need to include a point like point 2.2 (2.2. Review of empirical studies focusing on capital structure and CSR). It gives an idea of little art in the literature review.

Response: Comment accepted. The part 2.2 was excluded from the revised version of the paper (thus, the numbering was adjusted). 

Point 4: Research hypotheses should be included in the literature review. This is because the hypotheses derive from the literature review. The context in which the study was conducted can be described. The method followed can be more explanatory.

Response: Comment partially accepted. Hypotheses are included into the literature review and the subchapter 3.1 and 2.2 (originally 2.3) are partially merged. The part dedicated to methods for verification of hypotheses is left in chapter 3 (in a "new" separate subchapter). The method used is described in more detail. 

Point 5: It does not make sense to put together the presentation of results and the discussion of results. They are two distinct sections. Figure 1 is not appropriate in the discussion of results.

Response: According to our experience, results and discussion parts can be separated or merged into one chapter. Both approaches make sense for us. Nevertheless, if the reviewer insists on separation, it can be done. Figure 1 is connected with verification of hypothesis H1. Therefore, it is quite important for the context of the paper.

Point 6: The contributions of the study to the literature and to practitioners should be emphasized.

Response: Comment accepted. The part with conclusions was extended by emphasizing theoretical as well as practical contribution of the paper.